# Molecular Implications of *ADIPOQ, GAS5, GATA4*, and *YAP1* Methylation in Triple-Negative Breast Cancer Prognosis

**DOI:** 10.3390/ijms262110652

**Published:** 2025-11-01

**Authors:** Mateusz Wichtowski, Agnieszka Kołacińska-Wow, Katarzyna Skrzypek, Ewa Jabłońska, Katarzyna Płoszka, Damian Kołat, Sylwia Paszek, Izabela Zawlik, Elżbieta Płuciennik, Natalia Potocka, Wojciech Fendler, Paweł Kurzawa, Paweł Bigos, Łukasz Urbański, Paulina Gibowska-Maruniak, Thomas Wow

**Affiliations:** 1Department of Surgical Oncology, Institute of Oncology, Poznan University of Medical Sciences Szamarzewskiego 84, 60-569 Poznan, Poland; pawel.bigos@usk.poznan.pl (P.B.); lukasz.urbanski@usk.poznan.pl (Ł.U.); paulina.gibowska@usk.poznan.pl (P.G.-M.); 2Department of Oncological Physiotherapy, Medical University of Lodz, Paderewskiego 4, 93-509 Lodz, Poland; agnieszka.kolacinska@umed.lodz.pl; 3Department of General, Gastroenterological and Oncological Surgery, Warsaw Medical University, Banacha 1a, 02-097 Warsaw, Poland; 4Department of Translational Research, Nofer Institute of Occupational Medicine, St. Teresy 8 Street, 91-348 Lodz, Poland; kateryna.tarhonska@imp.lodz.pl; 5Department of Chemical Safety, Nofer Institute of Occupational Medicine, St. Teresy 8 Street, 91-348 Lodz, Poland; ewa.jablonska@imp.lodz.pl; 6Department of Biostatistics and Translational Medicine, Medical University of Lodz, Mazowiecka 15, 92-215 Lodz, Poland; katarzyna.ploszka@umed.lodz.pl (K.P.); wojciech.fendler@umed.lodz.pl (W.F.); 7Department of Experimental Surgery, Faculty of Medicine, Medical University of Lodz, 90-136 Lodz, Poland; damian.kolat@umed.lodz.pl; 8Department of Functional Genomics, Medical University of Lodz, Zeligowskiego 7/9, 90-752 Lodz, Poland; elzbieta.pluciennik@umed.lodz.pl; 9Department of General Genetics, Faculty of Medicine, Collegium Medicum, University of Rzeszow, 35-310 Rzeszow, Poland; sylwia.paszek@wp.pl (S.P.); izazawlik@gmail.com (I.Z.); 10Laboratory of Molecular Biology, Centre for Innovative Research in Medical and Natural Sciences, Faculty of Medicine, Collegium Medicum, University of Rzeszow, 35-959 Rzeszow, Poland; npotocka@ur.edu.pl; 11Department of Oncological Pathology, University Clinical Hospital in Poznan, Poznan University of Medical Sciences, 60-514 Poznań, Poland; pawel.kurzawa@usk.poznan.pl; 12Medical Practice Thomas Wow, 53 Malwowa Street, 60-175 Poznan, Poland; doctor.thomaswow@gmail.com

**Keywords:** TNBC, *ADIPOQ*, *GAS5*, *GATA4*, *YAP1*

## Abstract

The aim of this study was to investigate the prognostic and predictive properties of four specific genes in triple-negative breast cancer (TNBC). We focused on *ADIPOQ, GAS5, GATA4*, and *YAP1*, which are known for their roles in key molecular pathways related to tumorigenesis, such as adipokine signaling, lncRNA regulation, transcriptional control, and Hippo signaling, but have not been sufficiently explored in the context of epigenetic regulation in breast cancer. Using the methylospecific PCR (MSP) method, we analyzed the methylation of the four genes in the tumor tissues collected from 57 TNBC patients. We evaluated their association with response to neoadjuvant treatment and clinicopathological characteristics. Additionally, we performed a bioinformatic analysis of methylation and expression data from The Cancer Genome Atlas (TCGA) TNBC cohort to explore their relationships with overall survival (OS), disease-specific survival (DSS), disease-free interval (DFI), progression-free interval (PFI), and relapse-free survival (RFS). No significant associations were observed between methylation patterns and clinicopathological characteristics in the patients. However, in silico analysis of the TNBC cohort identified *ADIPOQ* methylation as having the most significant associations, correlating with all five survival endpoints, including OS, DSS, DFI, PFI, and RFS. *GAS5* methylation was significantly associated with OS, DSS, and RFS, and *GATA4* methylation showed significant associations with PFI, whereas *YAP1* methylation was significantly associated with OS and RFS. In addition, *GAS5* expression was linked to DSS, DFI and RFS. This study highlights the potential prognostic significance of the epigenetic regulation of *ADIPOQ* in TNBC. The in silico findings shed light on the molecular pathways associated with TNBC progression and warrant further investigation to validate their role in clinical outcomes and underlying biological mechanisms.

## 1. Introduction

Breast cancer ranks as the second most frequently diagnosed malignancy globally and remains the most common cancer among women [1,2]. Triple-negative breast cancer (TNBC), defined by the absence of estrogen receptor (ER), progesterone receptor (PR), and human epidermal growth factor receptor 2 (HER2) expression, represents a particularly aggressive and heterogeneous subtype. Accounting for approximately 15–20% of all breast cancer cases, TNBC is associated with a poor prognosis, elevated rates of recurrence, and distant metastasis [3]. The lack of targeted therapies means that cytotoxic chemotherapy remains the cornerstone of treatment. However, the development of resistance is common, highlighting the critical need to identify novel biomarkers that can inform prognosis and predict treatment response to improve patient outcomes [4].

Recent research has underscored the significant contribution of epigenetic modifications to the development and progression of TNBC. Among these, DNA methylation—the addition of a methyl group to cytosine bases in CpG islands, often mediated by DNA methyltransferases (DNMTs)—is a key mechanism for the transcriptional silencing of tumor suppressor genes [5]. In TNBC, while the total number of methylated CpG islands is comparable to other subtypes, the specific genes affected are distinct, influencing crucial processes such as proliferation, migration, and epithelial–mesenchymal transition (EMT) [6]. These alterations contribute to the aggressive phenotype of TNBC, making the profiling of methylation events a promising strategy for uncovering new biological insights and clinical biomarkers.

In this study, we focus on the epigenetic regulation of four genes—*ADIPOQ*, *GAS5*, *GATA4*, and *YAP1*—selected for their established roles in fundamental cancer pathways and their potential for methylation-mediated dysregulation in TNBC.

*ADIPOQ* encodes adiponectin, a hormone predominantly secreted by adipose tissue that regulates glucose metabolism and insulin sensitivity. Beyond its metabolic functions, adiponectin possesses significant anti-inflammatory, antiangiogenic, and antitumor properties [7]. Its actions are mediated through receptors AdipoR1 and AdipoR2, activating pathways like AMPK and PPAR-α, which are involved in cell growth, apoptosis, and inflammation [8]. In breast cancer, lower levels of adiponectin are associated with an increased risk of the disease, and it can inhibit cancer cell proliferation by suppressing oncogenic pathways such as PI3K/AKT and mTOR [9,10]. Therefore, the aberrant methylation and subsequent silencing of *ADIPOQ* could facilitate tumor development and therapy resistance, positioning it as a compelling candidate for investigation.

*GAS5* is a long non-coding RNA (lncRNA) that functions as a tumor suppressor by regulating cell cycle progression and apoptosis. It exerts its effects by acting as a molecular sponge for microRNAs (miRNAs); for instance, it sequesters miR-21 to inhibit mTOR signaling, a pathway crucial for cell proliferation and survival [11,12]. The downregulation of *GAS5* is implicated in enhanced tumor aggressiveness. Notably, epigenetic silencing of *GAS5* via promoter hypermethylation has been documented in other cancers, such as colorectal and non-small cell lung cancer [11], suggesting its potential utility as an epigenetic biomarker in TNBC.

*GATA4* is a zinc-finger transcription factor vital for cell differentiation, proliferation, and survival in various tissues [13]. In breast cancer, *GATA4* has been shown to act as a tumor suppressor by inhibiting invasion and migration, partly through the downregulation of *MMP9* expression [14]. Its role in regulating epithelial–mesenchymal transition (EMT) positions it as a key player in metastasis. The silencing of such a transcription factor via promoter methylation could thus be a pivotal event in TNBC pathogenesis.

*YAP1* serves as the principal downstream effector of the evolutionarily conserved Hippo signaling pathway, which regulates organ size, tissue homeostasis, and carcinogenesis [15]. When the Hippo pathway is inactive, *YAP1* translocates to the nucleus and partners with transcription factors like TEAD to drive the expression of pro-proliferative and antiapoptotic genes [16]. *YAP1* is frequently overexpressed in cancers and is known to promote tumorigenesis, but its role is complex and context-dependent, with evidence supporting both oncogenic and tumor-suppressive functions in different breast cancer models [17,18,19]. In TNBC, where Hippo pathway dysregulation is common, *YAP1* activation is thought to contribute to tumor progression and therapeutic resistance, making its regulatory mechanisms an area of intense interest [20].

The selection of *ADIPOQ*, *GAS5*, *GATA4*, and *YAP1* was based on their involvement in critical biological processes and evidence of their epigenetic regulation in other malignancies. However, their promoter methylation status and its clinical implications in TNBC remain insufficiently explored. We hypothesized that investigating the methylation of these biologically relevant genes could reveal novel prognostic biomarkers.

This study aims to analyze the correlation between the promoter methylation patterns of *ADIPOQ, GAS5, GATA4*, and *YAP1* and clinicopathological features, response to neoadjuvant chemotherapy, and survival outcomes in TNBC patients. By integrating data from a clinical cohort and a validation in-silico analysis of the TCGA dataset, we seek to clarify their prognostic and predictive significance, ultimately contributing to the development of more effective strategies for this challenging disease.

## 2. Results

Significant age differences were observed between subgroups: NAC patients were younger (mean age 52.1 ± 12.2 years, median 49.5) compared to surgery-first patients (mean 61.4 ± 15.8 years, median 62; *p* < 0.05). All tumors exhibited estrogen receptor (ER), progesterone receptor (PR), and human epidermal growth factor receptor 2 (HER2) negativity. High proliferative activity was evident across the group (median Ki-67: 60%, range: 8–90%), with no significant Ki-67 difference between subgroups (*p* > 0.05).

Histopathological evaluation revealed invasive carcinoma of no special type (NST/NOS) in 82.5% of cases, while grade 3 tumors predominated (56.1%). NAC patients presented with more advanced local disease, including higher frequencies of T3/T4 tumors (40.0% vs. 10.8%) and N2/N3 nodal involvement (30.0% vs. 8.1%). Stage IIB–III disease was common in both groups (50.0% NAC vs. 40.5% surgery-first). Pathological features included vascular emboli (31.6%), multifocality (14.0%), and skin invasion (12.3%). Pathological complete response (pCR) to NAC was achieved in 20.0% of patients, with partial response in 40.0% and no response/progression in 40.0%.

Methylation results of laboratory analysis are presented in Table 1, as qualitative data categorized into three groups (unmethylated, partially methylated, and methylated). The *GAS5* gene was found to be unmethylated in 95% of the samples, while the remaining 5% could not be analyzed due to technical problems. Therefore, it was excluded from further analysis. For the other three genes (*ADIPOQ*, *GATA4*, and *YAP1)*, the methylation patterns varied across the samples. Still, they were not associated with clinicopathological characteristics, including *Ki-67* (Figure 1), tumor grading (G), tumor staging (T), and lymph node metastasis (N) (Table 2). Similarly, no association was observed between methylation status and response to neoadjuvant chemotherapy (Table 3). For *ADIPOQ*, the OR for achieving pCR in methylated versus unmethylated samples was 1.67 (95% CI: 0.19–25.50; *p* > 0.999). For *YAP1*, the OR was 2.75 (95% CI: 0.13–32.04; *p* = 0.489). For *GATA4*, the OR could not be reliably calculated due to the absence of pCR events in the unmethylated subgroup. Although the observed trends suggested a potential increase in the likelihood of pCR in methylated tumors, none of these associations reached statistical significance.

However, a statistically significant association was observed between the methylation of *ADIPOQ* and *YAP1* genes and the age at diagnosis (Figure 2). Specifically, women diagnosed at a younger age had significantly higher methylation level of *ADIPOQ* in the tumor tissue compared to older patients (53 vs. 62 years, *p* = 0.037, Figure 2a). In contrast, a younger age at diagnosis was associated with a lack of methylation of *YAP1* (54 vs. 63 years, *p* = 0.034, Figure 2c).

### In Silico Group

The TCGA TNBC cohort included 172 patients with a median age of 54 years, with a slightly higher proportion of patients over 50 (54.7%). The racial distribution shows a significant representation of Black or African American patients (39%). The majority of patients (57%) were diagnosed at Stage II, followed by Stage III (26.2%). Invasive Ductal Carcinoma (IDC) is the predominant histological type (83.1%). The cohort includes rarer but clinically significant TNBC variants, such as Metaplastic Carcinoma (5.8%). Table 4 provides a comprehensive overview of the patient population, which is essential for interpreting subsequent genomic or survival analyses within this cohort.

Results of the in silico analysis of the TCGA TNBC cohort are summarized in Table 5. In accordance with our pre-specified plan, we considered disease-specific survival (DSS) as the primary outcome. Overall survival (OS), disease-free interval (DFI), progression-free interval (PFI), and relapse-free survival (RFS) were analyzed as secondary endpoints. This analysis identified *ADIPOQ* methylation as having the most significant associations with all five survival endpoints, including OS (*p* = 0.028), DSS (*p* = 0.023), DFI (*p* = 0.013), PFI (*p* = 0.037), and RFS (*p* = 0.011) (Figure 3). For all these endpoints, higher *ADPOQ* methylation was associated with better survival outcomes (HR = 0.422, HR = 0.342, HR = 0.340, HR = 0.459, HR = 0.304, respectively). Similarly, higher *GATA4* methylation showed a significant association with longer PFI (HR = 0.253, *p* = 0.044). In contrast, higher *GAS5* methylation was significantly associated with shorter OS (HR = 3.000, *p* = 0.005), DSS (HR = 3.350, *p* = 0.009), and RFS (HR = 3.160, *p* = 0.009). Higher *YAP1* methylation was associated with longer RFS (HR = 0.342, *p* = 0.031) but shorter OS (HR = 2.400, *p* = 0.049), although the latter might be inconclusive due to the *p*-value on the borderline of statistical significance. In addition, higher *GAS5* expression was linked to shorter DSS (HR = 3.200, *p* = 0.049), DFI (HR = 2.500, *p* = 0.048) and RFS (HR = 7.940, *p* = 0.016).

## 3. Discussion

The results of this study highlight the potential role of epigenetic regulation of *ADIPOQ*, *GAS5*, *GATA4*, and *YAP1* in the molecular pathology of triple-negative breast cancer. While our clinical cohort was limited in size, the in silico analysis of the larger TCGA TNBC cohort provided evidence supporting the role of these genes in survival outcomes. The lack of significant associations in our clinical group is likely attributable to its smaller sample size, which limits statistical power for survival analysis, and the semi-quantitative nature of MSP compared to the quantitative methylation arrays used in TCGA. The integration of clinical and bioinformatic data allowed us to explore these relationships, offering valuable insights. The in silico analysis of the TCGA TNBC cohort revealed significant associations between the methylation of *ADIPOQ*, *GAS5*, *GATA4*, and *YAP1* and various survival endpoints, including our primary endpoints: disease-specific survival, and secondary: overall survival, disease-free interval, progression-free interval, and relapse-free survival.

*ADIPOQ* methylation emerged as the most significant predictor of survival outcomes, correlating with all five survival endpoints in the in silico study. Higher *ADIPOQ* methylation was associated with improved OS, DSS, DFI, PFI, and RFS. Interestingly, no correlation was observed with gene expression, although lower expression would be expected in cases of higher methylation. In our TNBC samples, we observed significantly higher *ADIPOQ* methylation in younger patients; however, no correlations were found with clinicopathological characteristics or the Ki-67 index. This contrasts with most studies, which have shown that low adiponectin levels are associated with increased risk and severity of breast cancer [21,22,23]. Our inconsistent findings may be due to certain limitations, such as the small sample size, limited data, and the specific method used for epigenetic analysis.

In contrast to *ADIPOQ*, higher *GAS5* methylation was associated with poorer OS, DSS, and RFS. Observations from previous studies indicate that CpG methylation in the promoter region of the *GAS5* gene is increased in TNBC tissues and cell lines. At the same time, *GAS5* expression is suppressed in these tumor cells. Interestingly, we also observed that higher *GAS5* expression was linked to shorter DSS, DFI, and RFS, even though hypermethylation of *GAS5* is typically expected to silence its expression, not increase it. This observation is inconsistent with previous data that demonstrated the tumor-suppressive function of *GAS5*. According to the UCSC Human Genome Browser (GRCh38/hg38; accessed in March 2023) [24], the *GAS5* gene is located on human chromosome 1q25.1 (chr1:173, 858, 997–173, 867, 989), and a CpG island has been identified immediately upstream of the *GAS5* gene (chr1:173, 868, 035–173, 868, 779) [25]. The sequence analyzed in our study (chr1:173, 868, 568–173, 868, 695) is located within this CpG island. The lack of methylation observed in this region may result from the fact that CpG islands within promoter regions often exhibit heterogeneous methylation pattern-different parts of the same island can vary in the degree of CpG methylation. Moreover, the MSP (methylation-specific PCR) technique is locus-specific, meaning that it detects methylation only at CpG sites covered by the primers. Even slight shifts in primer location compared with other studies [26,27] may lead to the analysis of distinct fragments of the CpG island and, consequently, to apparently divergent results.

In summary, the differences between our results and those reported by other authors most likely arise from variations in the position of the analyzed amplicons within the CpG island or from methodological differences in methylation detection. It should also be emphasized that the absence of methylation in the examined fragment does not exclude the tumor-suppressive role of *GAS5*, since its expression may also be regulated through methylation of other promoter regions or through alternative epigenetic mechanisms (e.g., histone modifications or microRNA-mediated regulation).

Through epigenetic and other regulatory mechanisms, *GAS5* has been shown to enhance drug sensitivity, improve prognosis, and promote apoptosis in breast cancer [28]. This paradoxical observation suggests that the role of *GAS5* may be more complex than previously understood, potentially involving isoform-specific effects or context-dependent functions that warrant further investigation.

*GATA4* and *YAP1* methylation showed more limited but still significant associations with survival outcomes. In our bioinformatics analysis, higher *GATA4* methylation was associated with longer PFI, which contrasts with findings from other studies reporting that *GATA4* suppresses cell proliferation, invasion, and migration, while promoting apoptosis and senescence in breast cancer cells which lacks hormone receptors and HER2, *GATA4* may interact with different transcriptional networks, possibly resulting in context-dependent effects, including pro-tumor functions when overexpressed [29,30].

On the other hand, *YAP1* methylation was associated with more prolonged RFS but shorter OS, indicating a complex and context-dependent role in TNBC biology. *YAP1* functions in both tumor suppression and promotion depending on the cellular environment, co-regulators, and stage of disease [18,31]. These results highlight the need to dissect the isoform-specific, spatial, and temporal regulation of *YAP1* in TNBC progression.

The in silico results demonstrate the power of bioinformatic analysis in identifying potential biomarkers and uncovering molecular pathways relevant to TNBC. However, it is essential to acknowledge the limitations of our study group, which was small and may not fully capture the heterogeneity of TNBC.

In our study, we observed a significant association between the methylation status of *ADIPOQ* and the age at diagnosis. Specifically, younger patients (median age: 53 years) had higher *ADIPOQ* methylation levels compared to older patients (median age: 62 years). This finding suggests that *ADIPOQ* methylation may play a role in the early onset of TNBC, potentially contributing to the aggressive behavior of tumors in younger patients [32]. Given that high methylation corresponds to low expression, the hypermethylation of *ADIPOQ* in younger patients may result in reduced adiponectin levels, which could promote tumor growth and metastasis [7]. Conversely, younger patients (median age: 54 years) showed a lack of *YAP1* methylation compared to older patients (median age: 63 years). This finding aligns with the known oncogenic role of *YAP1*, which promotes cell proliferation and survival [32]. The absence of *YAP1* methylation in younger patients may lead to higher *YAP1* expression, contributing to the aggressive behavior of TNBC in this age group. These age-related differences in methylation patterns highlight the potential role of epigenetic regulation in the biology of TNBC and suggest that age-specific molecular mechanisms may influence tumor behavior and patient outcomes. Further studies are needed to explore the underlying mechanisms and validate these findings in larger cohorts.

In the subgroup of patients treated with neoadjuvant chemotherapy in our study group, we did not observe statistically significant associations between the methylation status of the analyzed genes (*ADIPOQ*, *GATA4*, and *YAP1*) and pathological complete response. Although higher OR values for *ADIPOQ* and *YAP1* suggested a potential trend toward increased likelihood of pCR in methylated tumors, the wide confidence intervals reflect considerable uncertainty due to the small sample size.

While this study provides valuable insights, several limitations must be acknowledged. First, the small sample size of our study group limits the statistical power of our findings, particularly within the neoadjuvant chemotherapy subgroup. Therefore, these results should be interpreted with caution and regarded as exploratory, warranting confirmation in larger, independent cohorts with quantitative methylation assessment and parallel gene expression profiling. Furthermore, the associations reported are based on univariable analysis and were not adjusted for potential confounders such as age or disease stage. Therefore, these findings are exploratory and must be validated in future, dedicated studies with pre-defined cohorts and multivariable analysis (our preliminary multivariate analysis in Appendix A). Second, the in silico analysis, while insightful, must be interpreted with caution. The use of data-driven optimal cut-points and the analysis of multiple survival endpoints increase the risk of false-positive findings. We attempted to mitigate this by pre-specifying a primary outcome (DSS); however, we did not apply corrections for multiple testing (e.g., FDR) to maintain sensitivity for this hypothesis-generating analysis. Consequently, the significant associations reported, particularly for the secondary endpoints, are exploratory and must be validated in future, dedicated studies with pre-defined, biologically justified cut-points and appropriate statistical corrections. Third, the use of archival FFPE samples and MSP may introduce variability compared to quantitative techniques. Fourth, the in silico analysis, while robust, is based on retrospective data. Finally, the technical failure to analyze *GAS5* methylation in a small subset of samples underscores the challenges of working with FFPE-derived DNA. Our conservative approach to exclude these samples ensured data integrity. Future studies should include larger, prospective cohorts using quantitative methylation assays (e.g., pyrosequencing) and integrate matched gene expression data to link methylation status to biological outcomes functionally. Mechanistic studies are also needed to explore the causal roles of these genes.

In conclusion, this study highlights the potential prognostic significance of *ADIPOQ*, *GAS5*, *GATA4*, and *YAP1* methylation in TNBC. The in silico analysis of the TCGA cohort provided strong evidence, particularly for *ADIPOQ* methylation as a high-priority candidate biomarker, while the clinical findings revealed age-related differences. This work serves as a discovery and hypothesis-generating study. The robust association of *ADIPOQ* methylation with survival in a large independent dataset justifies its prioritization for further validation in large, prospective TNBC cohorts to assess its true clinical utility. Future work should include functional studies to elucidate the precise molecular mechanisms by which methylation of these genes influences TNBC cell behavior.

## 4. Materials and Methods

### 4.1. Study Group

The study group comprised 57 patients with histologically confirmed triple-negative breast cancer (TNBC), including 20 patients (35.1%) who received neoadjuvant chemotherapy (NAC) and 37 patients (64.9%) treated with upfront surgery.

Study characteristics are presented in Table 6.

The study was conducted in accordance with the Declaration of Helsinki and approved by the Bioethics Committee at the Medical University of Lodz (RNN/226/11/KE). As the samples were archival and anonymized, individual informed consent was not required.

### 4.2. Gene Methylation Analysis

DNA was isolated from formalin-fixed paraffin-embedded (FFPE) tumor samples using the QIAamp DNA FFPE Tissue Kit (Qiagen, Hilden, Germany) and then modified by sodium bisulfate using the EpiTect Bisulfite Kit (Qiagen) according to the manufacturer’s protocols. The converted DNA was stored at −20 °C until further analysis. DNA methylation was analyzed by methyl-specific PCR (MSP). The promoter sequences of the *ADIPOQ*, *GAS5*, *GATA4*, and *YAP1* genes were assessed for methylation. The primer sequences for the *GAS5* gene were designed using Methyl Prime 2.0 software, and for the other genes were described previously [33,34,35,36]. Bisulfite-converted methylated and unmethylated human DNA from the EpiTect PCR Control DNA Set (Qiagen) was used as positive and negative controls. The MSP reaction mixture in a volume of 10 µL contained 1× Buffer, 2.5 mM MgCl_2_, 0.25 mM dNTPs, 0.25 U TaKaRa EpiTaq HS (Takara Bio, Shiga, Japan), and primers in the concentration range of 0.2–0.4 µm. The PCR reaction conditions were as follows: 10 s of initial denaturation at 98 °C, followed by 35 cycles of 20 s of denaturation at 94 °C, 30 s of annealing at a temperature depending on the primer, 30 s of extension at 72 °C, and a final extension at 72 °C for 7 min. Table 7 shows the annealing temperature, primer sequences, and the length of the PCR product.

Methylation status was presented as qualitative data, categorized into three groups (unmethylated, partially methylated, methylated) or two groups (unmethylated, partially methylated + methylated). The classification was based on the visual intensity of PCR bands after agarose gel electrophoresis: unmethylated, methylated, and partially methylated.

Samples were designated as methylated if a product was obtained only in the reaction with methylated primers and as unmethylated if a product was obtained only in the reaction with unmethylated primers. Partial methylation was considered if amplicons were observed on the gel in both MSP reactions (Figure 4). The main limitation was the biological material type, as FFPE-derived DNA is prone to fragmentation and degradation and may contain PCR inhibitors, factors that could have influenced amplification efficiency.

### 4.3. Bioinformatic Analysis

The mRNA expression and DNA methylation data were acquired from the University of California, Santa Cruz (UCSC) “Xena” repository [25] for TNBC samples of The Cancer Genome Atlas (TCGA)-BRCA cohort [37,38]. Clinical data were collected from UCSC Xena and cBioPortal, which included overall survival (OS), disease-specific survival (DSS), disease-free interval (DFI), progression-free interval (PFI), and relapse-free survival (RFS) [39,40]. For the bioinformatic survival analysis, disease-specific survival (DSS) was pre-specified as the primary endpoint. Overall survival (OS), disease-free interval (DFI), progression-free interval (PFI), and relapse-free survival (RFS) were considered secondary, exploratory endpoints. Survival analysis was performed in RStudio v2023.03.0+386 (R version 4.2.3) using the survminer v0.4.9 package to determine an optimal cutpoint that best segregated patients into high and low methylation/expression groups based on survival outcome. The hazard ratio (HR) was calculated for the group with higher values relative to the group with lower values.

### 4.4. Statistical Analysis

Statistical analysis of laboratory and clinical data was conducted via STATISTICA version. 13.3 (TIBCO Stat software Inc., Palo Alto, CA, USA). Differences between continuous variables (age, Ki-67) concerning methylation status (divided into two or three categories) were analyzed using Student’s *t*-test or ANOVA. For non-normally distributed data, the Mann–Whitney U test and the Kruskal–Wallis test were applied. Associations between qualitative clinicopathological data and methylation status, as well as between chemotherapy response and methylation status (2 × 2 contingency table analysis), were evaluated with Fisher’s exact test. In the NAC patients, odds ratios (ORs) with 95% confidence intervals (CIs) were additionally calculated to estimate the effect size of associations between gene methylation and pCR. These calculations were performed in GraphPad Prism version. 7.04 (San Diego, CA, USA) using Fisher’s exact test and the Baptista–Pike method, which provides accurate CI estimates for small sample sizes. Data normality was assessed with the Shapiro–Wilk test, and the homogeneity of variances was evaluated using Levene’s test and Brown–Forsythe test. A *p*-value of 0.05 or lower was considered statistically significant. Quantitative data are presented in figures as box-and-whisker plots, displaying the median, interquartile range (IQR), and individual data points. Figures were created in GraphPad Prism version. 7.04 (San Diego, CA, USA).

## Figures and Tables

**Figure 1 ijms-26-10652-f001:**
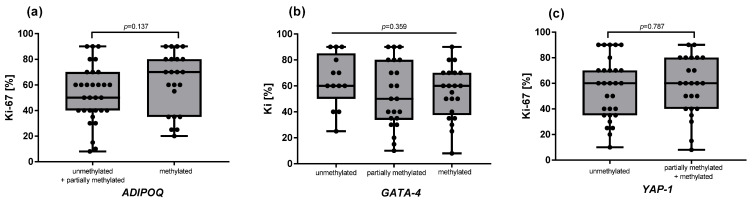
Ki-67 stratified by methylation status of *ADIPOQ* (**a**), *GATA-4* (**b**), and *YAP1* (**c**) in the tumor tissue. For the *ADIPOQ* gene (**a**), the “unmethylated” group was combined with the “partially methylated” group due to the presence of only one sample with an unmethylated status. For the *YAP-1* gene (**c**), the “partially methylated” group was combined with the “methylated” group due to the presence of only two samples with methylated status. Data are shown as raw values with medians and IQR. Group differences were analyzed with Student’s *t*-test (**c**), Mann–Whitney U-test (**a**), and one-way ANOVA (**b**).

**Figure 2 ijms-26-10652-f002:**
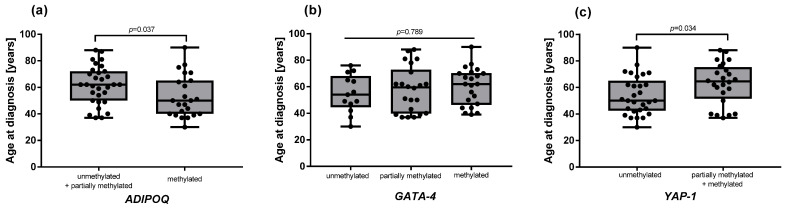
Age at diagnosis stratified by methylation status of *ADIPOQ* (**a**), *GATA-4* (**b**), and *YAP1* (**c**). For the *ADIPOQ* gene (**a**), the “unmethylated” group was combined with the “partially methylated” group due to the presence of only one sample with an unmethylated status. For the *YAP-1* gene (**c**), the “partially methylated” group was combined with the “methylated” group due to the presence of only two samples with methylated status. Data are shown as raw values with medians and IQR. Group differences were analyzed with Student’s *t*-test (**a**,**c**) and Kruskal–Wallis test (**b**).

**Figure 3 ijms-26-10652-f003:**
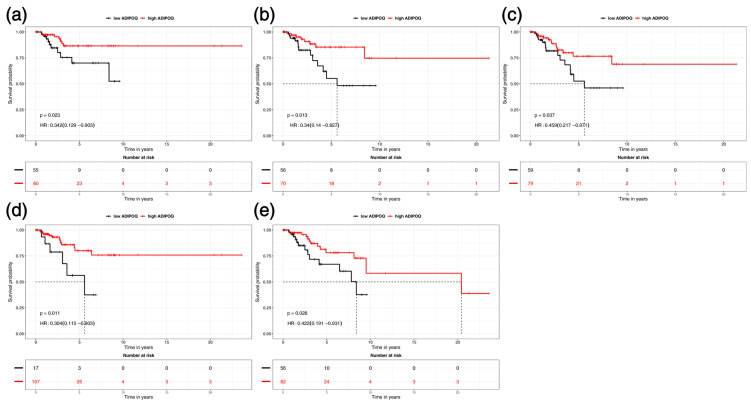
*ADIPOQ* methylation as having the most significant associations, being associated with all five survival endpoints in TCGA TNBC cohort, including (**a**) DSS (*p* = 0.023), (**b**) DFI (*p* = 0.013), (**c**) PFI (*p* = 0.037), (**d**) RFS (*p* = 0.011) and (**e**) OS (*p* = 0.028).

**Figure 4 ijms-26-10652-f004:**
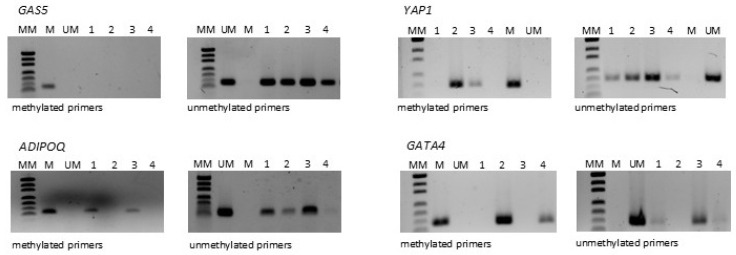
Representative products of MSP reaction. Abbreviations: MM—mass marker; M—methylated control (completely methylated DNA); UM—unmethylated control (completely unmethylated DNA); lanes 1–4 correspond to the analyzed samples.

**Table 1 ijms-26-10652-t001:** Methylation status of *ADIPOQ*, *GATA-4*, *GAS5*, and *YAP-1* genes in the tumor samples of TNBC patients.

Methylation Status	*ADIPOQ* *n* (%)	*GAS5* *n* (%)	*GATA4* *n* (%)	*YAP1* *n* (%)
All	57 (100.0%)	57 (100.0%)	57 (100.0%)	57 (100.0%)
Unmethylated	1 (1.8%)	54 (94.7%)	13 (22.8%)	29 (50.9%)
Partially Methylated	30 (52.6%)	0 (0.0%)	22 (38.6%)	22 (38.6%)
Methylated	23 (40.4%)	0 (0.0%)	15 (26.3%)	2 (3.5%)
Missing Data	3 (5.3%)	3 (5.3%)	7 (12.3%)	4 (7.0%)
**Biopsies**				
All	20 (100.0%)	20 (100.0%)	20 (100.0%)	20 (100.0%)
Unmethylated	0 (0.0%)	18 (90.0%)	5 (25.0%)	13 (65.0%)
Partially Methylated	6 (30.0%)	(0.0%)	2 (10.0%)	2 (10.0%)
Methylated	12 (60.0%)	(0.0%)	9 (45.0%)	1 (5.0%)
Missing Data	2 (10.0%)	2 (10.0%)	4 (20.0%)	4 (20.0%)
**Surgical Samples**				
All	37 (100.0%)	37 (100.0%)	37 (100.0%)	37 (100.0%)
Unmethylated	1 (2.7%)	36 (97.3%)	8 (21.6%)	16 (43.2%)
Partially Methylated	24 (64.9%)	0 (0.0%)	20 (54.1%)	20 (54.0%)
Methylated	11 (29.7%)	0 (0.0%)	6 (16.2%)	1 (2.7%)
Missing Data	1 (2.7%)	1 (2.7%)	3 (8.1%)	0 (0.0%)

**Table 2 ijms-26-10652-t002:** Clinicopathological characteristics of TNBC patients according to methylation status of *ADIPOQ*, *GATA-4*, and *YAP1* in the tumor tissue (2 × 2 contingency table analysis).

Clinicopathological Characteristics	*ADIPOQ* *n* = 54		*GATA4* *n* = 50		*YAP1* *n* = 53	
	Unmethylated + Partially Methylated (*n*)	Methylated (*n*)	Unmethylated (*n*)	Partially Methylated + Methylated (*n*)	Unmethylated (*n*)	Partially Methylated + Methylated (*n*)
Tumor Grading (G)						
G1 + G2	12	11	5	16	16	8
G3	19	12	8	21	13	16
*p* *	0.583		1.000		0.166	
Tumor Staging (T)						
T1 + T2	23	21	10	29	24	18
T3 + T4	8	2	3	8	5	6
*p* *	0.161		1.000		0.517	
Lymph Node Metastasis (N)						
N0	17	13	6	21	15	13
N1–3	14	10	7	16	14	11
*p **	1.000		0.537		1.000	

For the 2 × 2 contingency analysis, methylation status was categorized into two groups, distinguishing between unmethylated and either methylated or partially methylated samples. The exception was the *ADIPOQ* gene, in which case the “unmethylated” group was combined with the “partially methylated” group due to the presence of only one sample with unmethylated status. * Fisher’s exact test.

**Table 3 ijms-26-10652-t003:** Response to neoadjuvant chemotherapy in TNBC patients according to the methylation status of *ADIPOQ*, *GATA-4*, and *YAP1* in the tumor tissue collected before treatment (2 × 2 contingency table analysis).

Pathological Complete Response (pCR)	*ADIPOQ* *n* = 18		*GATA4* *n* = 16		*YAP1* *n* = 16	
	Methylated (*n*)	Unmethylated + Partially Methylated (*n*)	Partially Methylated + Methylated (*n*)	Unmethylated (*n*)	Partially Methylated + Methylated (*n*)	Unmethylated (*n*)
Yes	3	1	3	0	1	2
No	9	5	8	5	2	11
*p **	>0.999		0.509		0.489	
OR (95% CI)	1.67 (0.19–25.50)		N/A (0.39-infinity)		2.75 (0.13–32.04)	

For the *ADIPOQ* gene, the “unmethylated” group was combined with the “partially methylated” group due to the presence of only one sample with an unmethylated status. For the YAP-1 gene, the “partially methylated” group was combined with the “methylated” group due to the presence of only two samples with methylated status. * Fisher’s exact test. ORs and 95% CIs were estimated using the Baptista–Pike method. N/A—Not avaliable

**Table 4 ijms-26-10652-t004:** Baseline clinical and pathological characteristics of the TCGA TNBC cohort (*N* = 172).

Characteristic	Category	Total Cohort (*N* = 172)	% or Median (Range)
**Age at Diagnosis (years)**			
	Median (Range)		54 (26–90)
	≤50	78	45.3%
	>50	94	54.7%
**Race**			
	White	95	55.2%
	Black or African American	67	39.0%
	Asian	5	2.9%
	Not Available/Not Evaluated	5	2.9%
**AJCC Pathologic Stage**			
	Stage I	21	12.2%
	Stage II (IIA/IIB)	98	57.0%
	Stage III (IIIA/IIIB/IIIC)	45	26.2%
	Stage IV	2	1.2%
	Stage X/Not Available	6	3.5%
**Histological Type**			
	Invasive Ductal Carcinoma (IDC)	143	83.1%
	Metaplastic Carcinoma	10	5.8%
	Invasive Lobular Carcinoma (ILC)	7	4.1%
	Mixed/Other Specified Types	12	7.0%
**Menopausal Status**			
	Premenopausal	36	20.9%
	Perimenopausal	9	5.2%
	Postmenopausal	109	63.4%
	Indeterminate/Not Available	18	10.5%
**Vital Status (Follow-up)**			
	Alive	133	77.3%
	Deceased	39	22.7%

AJCC: American Joint Committee on Cancer; IDC: invasive ductal carcinoma; ILC: invasive lobular carcinoma.

**Table 5 ijms-26-10652-t005:** Summary results of in silico analysis (TCGA TNBC cohort).

Survival Endpoint	*ADIPOQ*	*GAS5*	*GATA4*	*YAP1*
	Methylation (high vs. low)			
**Primary Outcomes**				
DSS	**HR = 0.342** **(0.129–0.903)** ***p* = 0.023**	**HR = 3.350** **(1.270–8.830)** ***p* = 0.009**	HR = 0.258 (0.034–1.950) *p* = 0.160	HR = 1.29 × 10^−8^ (0–Inf) *p* = 0.190
**Secondary Outcomes**				
OS	**HR = 0.422** **(0.191–0.931)** ***p* = 0.028**	**HR = 3.000** **(1.340–6.740)** ***p* = 0.005**	HR = 1.520 (0.693–3.320) *p* = 0.290	**HR = 2.400** **(0.978–5.860)** ***p* = 0.049**
DFI	**HR = 0.340** **(0.140–0.827)** ***p* = 0.013**	HR = 0.443 (0.146–1.340) *p* = 0.140	HR = 0.375 (0.087–1.620) *p* = 0.170	HR = 0.455 (0.183–1.130) *p* = 0.082
PFI	**HR = 0.459** **(0.217–0.971)** ***p* = 0.037**	HR = 1.830 (0.865–3.890) *p* = 0.110	**HR = 0.253** **(0.059–1.070)** ***p* = 0.044**	HR = 0.536 (0.247–1.160) *p* = 0.110
RFS	**HR = 0.304** **(0.115–0.803)** ***p* = 0.011**	**HR = 3.160** **(1.280–7.820)** ***p* = 0.009**	HR = 0.555 (0.223–1.380) *p* = 0.200	**HR = 0.342** **(0.123–0.951)** ***p* = 0.031**
	Expression (high vs. low)			
**Primary Outcomes**				
DSS	HR = 1.710 (0.726–4.040) *p* = 0.214	**HR = 3.200** **(0.942–10.900)** ***p* = 0.049**	HR = 1.28 × 10^−8^ (0–Inf) *p* = 0.124	HR = 1.970 (0.828–4.680) *p* = 0.118
**Secondary Outcomes**				
OS	HR = 1.870 (0.852–4.100) *p* = 0.113	HR = 2.280 (0.798–6.540) *p* = 0.113	HR = 1.850 (0.909–3.770) *p* = 0.085	HR = 0.197 (0.027–1.450) *p* = 0.076
DFI	HR = 1.970 (0.808–4.790) *p* = 0.129	**HR = 2.500** **(0.976–6.400)** ***p* = 0.048**	HR = 1.990 (0.867–4.560) *p* = 0.098	HR = 0.248 (0.033–1.840) *p* = 0.140
PFI	HR = 0.581 (0.287–1.180) *p* = 0.127	HR = 2.560 (0.900–7.270) *p* = 0.067	HR = 0.262 (0.036–1.920) *p* = 0.156	HR = 0.355 (0.085–1.480) *p* = 0.137
RFS	HR = 0.541 (0.227–1.290) *p* = 0.160	**HR = 7.940** **(1.070–59.000)** ***p* = 0.016**	HR = 1.760 (0.756–4.090) *p* = 0.184	HR = 0.245 (0.033–1.820) *p* = 0.136

OS—overall survival, DSS—disease-specific survival, DFI—disease-free interval, PFI—progression-free interval, RFS—relapse-free survival; *p*-values ≤ 0.05 and statistically significant HR values are in bold. DSS was pre-specified as the primary outcome. OS, DFI, PFI, and RFS are secondary, exploratory endpoints. The use of optimal cut-points and the analysis of multiple endpoints increase the exploratory nature of these findings, which require independent validation.

**Table 6 ijms-26-10652-t006:** Characteristics of the study group.

Characteristic	Total (*n* = 57)	Preoperative Chemotherapy (NAC *n* = 20)	No Preoperative Chemotherapy (*n* = 37)
**Age at diagnosis (years)**			
Mean ± SD	58.1 ± 15.2	52.1 ± 12.2	61.4 ± 15.8
Median (full range)	60 (30–90)	49.5 (30–75)	62 (27–90)
**Ki-67 Index (%)**			
Mean ± SD	58.2 ± 22.6	55.5 ± 19.5	55.6 ± 24.3
Median (full range)	60 (8–90)	60 (25–90)	60 (8–90)
**Receptor Status**			
ER/PR/HER2 negative	57 (100%)	20 (100%)	37 (100%)
**Histopathological Type**			
NST/NOS (no special type)	47 (82.5%)	18 (90.0%)	29 (78.4%)
Medullary (M)	3 (5.3%)	1 (5.0%)	2 (5.4%)
Apocrine (ACC)	2 (3.5%)	0 (0%)	2 (5.4%)
Other subtypes	5 (8.7%)	1 (5.0%)	4 (10.8%)
**Tumor grade (G)**			
G1	1 (1.8%)	0 (0%)	1 (2.7%)
G2	24 (42.1%)	11 (55.0%)	13 (35.1%)
G3	32 (56.1%)	9 (45.0%)	23 (62.2%)
**T**			
T1	12 (21.1%)	1 (5.0%)	11 (29.7%)
T2	33 (57.9%)	11 (55.0%)	22 (59.5%)
T3/T4	12 (21.0%)	8 (40.0%)	4 (10.8%)
**N**			
N0	31 (54.4%)	7 (35.0%)	24 (64.9%)
N1	17 (29.8%)	7 (35.0%)	10 (27.0%)
N2/N3	9 (15.8%)	6 (30.0%)	3 (8.1%)
**M**			
M0	55 (96.5%)	19 (95.0%)	36 (97.3%)
M1	2 (3.5%)	1 (5.0%)	1 (2.7%)
**Stage (NAC)**			
I–IIA	32 (56.1%)	10 (50.0%)	22 (59.5%)
IIB–III	25 (43.9%)	10 (50.0%)	15 (40.5%)
**Surgery Type**			
Breast-conserving (BCT)	25 (43.9%)	8 (40.0%)	17 (45.9%)
Mastectomy (M)	32 (56.1%)	12 (60.0%)	20 (54.1%)
**Pathological Features**			
Multifocality	8 (14.0%)	4 (20.0%)	4 (10.8%)
Vascular emboli	18 (31.6%)	7 (35.0%)	11 (29.7%)
Skin invasion	7 (12.3%)	2 (10.0%)	5 (13.5%)
Extensive intraductal (EIC)	2 (3.5%)	0 (0%)	2 (5.4%)

**Abbreviations**: SD—standard deviation; ER—estrogen receptor; PR—progesterone receptor; HER2—human epidermal growth factor receptor 2; NST/NOS—no special type; ACC—apocrine carcinoma;. NAC—neoadjuvant chemotherapy.

**Table 7 ijms-26-10652-t007:** Annealing temperature, primer sequences, and MSP reaction product size.

Gene	Primer Sequence	Product Size [bp]	Annealing Temperature [°C]
*ADIPOQ* M	F: TATTTTATATGATTATATTTCGCGG R: AACTCTATTCTAACTTCCTAACGAA	121	57
*ADIPOQ* UM	F: TTTATTTTATATGATTATATTTTGTGG R: AACTCTATTCTAACTTCCTAACAAA	123	57
*GAS5* M	F: AGTTGTTAGGAGGTGGGTGTGC R: CCCGACCGAACTAATCTACC	128	63
*GAS5* UM	F: AGTTGTTAGGAGGTGGGTGTGT R: CTTAACCCCCAACCAAACTAATCTACC	135	63
*GATA4* M	F: GTATAGTTTCGTAGTTTGCGTTTAGC R: AACTCGCGACTCGAATCCCCG	136	65
*GATA4* UM	F: TTTGTATAGTTTTGTAGTTTGTGTTTAGT R: CCCAACTCACAACTCAAATCCCCA	142	62
*YAP1* M	F: AGTTCGTATAGGCGTTTCGTTC R: CTTAACTACAAAAAATTCTTCCGCT	187	57
*YAP1* UM	F: AAGTTTGTATAGGTGTTTTGTTTGG R: CTTAACTACAAAAAATTCTTCCACT	188	58

## Data Availability

The original contributions presented in this study are included in the article/Appendix A. Further inquiries can be directed to the corresponding author.

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
