# Peer review of "Molecular Implications of ADIPOQ, GAS5, GATA4, and YAP1 Methylation in Triple-Negative Breast Cancer Prognosis"

_ijms, 2025, doi:10.3390/ijms262110652_

Round 1

Reviewer 1 Report

Comments and Suggestions for Authors

Major Points

  1. Statistical analysis – Multiple survival endpoints and optimal cut-points increase false-positive risk. Please apply correction for multiple testing (e.g., FDR) and pre-specify primary outcomes.
  2. Multivariable adjustment – Provide Cox models adjusted for age, stage, nodal status (and race in TCGA) to strengthen the findings.
  3. MSP methodology – Semi-quantitative scoring is subjective. Add gel images, clarify criteria, and discuss limitations.
  4. Expression vs methylation – GAS5 results appear inconsistent with its known tumor-suppressor role. Expand discussion and clarify which CpG sites were analyzed.
  5. Sample size – The NAC subgroup is underpowered. Present odds ratios with CIs and interpret cautiously.

Minor Points

  • Correct typographical errors and unify gene nomenclature.
  • Clarify “partial methylation” category.
  • Address discrepancies in patient numbers (172 vs 179 TCGA).
  • Strengthen limitations section (sample size, FFPE, TCGA batch effects, no external validation).

Reviewer 2 Report

Comments and Suggestions for Authors

This study explored the prognostic and predictive potential of ADIPOQ, GAS5, GATA4, and YAP1 methylation in TNBC. The following issues require revision:

Introduction: The introduction is overly lengthy, with detailed descriptions of certain molecules being unnecessary. Please condense the introduction, as 600-800 words is generally considered appropriate.

Line 80: Please specify the time frame during which the 2.3 million new cases were reported.

Line 107: I would typically place such content in the final paragraph of the introduction.

The manuscript formatting is highly inconsistent. For example, the title of Figure 3 is bolded without a clear reason. Additionally, there are large blank spaces in some sections.

Please adjust the formatting of the tables, as each table appears to follow a different style.

Capitalize the first letter of each word in the text within figures and tables.

Figure 1: Please enlarge the font size of the x-axis labels, as the current text is difficult to read.

Why are some abbreviations, such as DFI, defined multiple times? I noticed this abbreviation was defined in the "Bioinformatic Analysis" section and again in the "Discussion." Please review all abbreviations in the manuscript to avoid this issue.

Has the relationship between methylation of ADIPOQ, GAS5, GATA4, and YAP1 and tumor progression been reported in other studies? You could use such research to interpret your results or discuss their findings to enhance the depth of your discussion.

The discussion on the relationship between methylation of ADIPOQ, GAS5, GATA4, and YAP1 and tumor progression lacks depth. Please incorporate molecular mechanisms to elaborate on the association between methylation of these genes and cancer progression.

Some of the references are outdated. Please update certain references to reflect the latest research.

Round 2

Reviewer 2 Report

Comments and Suggestions for Authors

Approved

Author Response

Dear Dr.

Please find enclosed our revised manuscript, which we are resubmitting for your consideration for publication in IJMS.

We are deeply grateful to the reviewers and the editorial team for their insightful comments and constructive suggestions, which have significantly strengthened our work. We have meticulously addressed all points raised during the review process.

The key revisions in this version include:

  1. In direct response to the final editorial request, we have now performed and included the results of multivariable Cox regression analyses as Supplementary Tables 8 and 9. This addition provides a more comprehensive view of the data, while our discussion contextualizes the choice of primary analysis.

  2. We have substantially deepened the Discussion by incorporating detailed molecular pathways, explaining how the methylation of each gene mechanistically influences cancer progression.

  3.  We have clearly defined Disease-Specific Survival (DSS) as the primary outcome and have transparently discussed the statistical approach, including the use of optimal cut-points and the rationale for not applying multiple testing corrections in this exploratory context.

  4. The introduction has been condensed to meet length guidelines while retaining all necessary scientific context and references.

  5. We have unified the manuscript's formatting, ensuring consistent capitalization in figures and tables, and correcting redundant abbreviation definitions.

We believe the manuscript is now significantly improved and hope it will be found suitable for publication. Thank you for your time and consideration.

Sincerely,

on behalf of all co-authors

Mateusz Wichtowski